# Intestinal Stem Cells Damaged by Deoxycholic Acid via AHR Pathway Contributes to Mucosal Barrier Dysfunction in High-Fat Feeding Mice

**DOI:** 10.3390/ijms232415578

**Published:** 2022-12-08

**Authors:** Leheng Liu, Jingxian Xu, Xianjun Xu, Tiancheng Mao, Wenlu Niu, Xiaowan Wu, Lungen Lu, Hui Zhou

**Affiliations:** 1Department of Gastroenterology, Shanghai General Hospital, Shanghai Jiao Tong University School of Medicine, Shanghai 200080, China; 2Shanghai Key Laboratory of Pancreatic Diseases, Shanghai Jiao Tong University School of Medicine, Shanghai 201620, China

**Keywords:** high-fat diet, deoxycholic acid, aryl hydrocarbon receptor, intestinal stem cells, kynurenine, indoleamine 2,3-dioxygenase 1

## Abstract

High-fat exposure leads to impaired intestinal barrier function by disrupting the function of intestinal stem cells (ISCs); however, the exact mechanism of this phenomenon is still not known. We hypothesize that high concentrations of deoxycholic acid (DCA) in response to a high-fat diet (HFD) affect aryl hydrocarbon receptor (AHR) signalling in ISCs and the intestinal barrier. For this purpose, C57BL/6J mice feeding on a low-fat diet (LFD), an HFD, an HFD with the bile acid binder cholestyramine, and a LFD with the DCA were studied. We found that high-fat feeding induced an increase in faecal DCA concentrations. An HFD or DCA diet disrupted the differentiation function of ISCs by downregulating AHR signalling, which resulted in decreased goblet cells (GCs) and MUC2, and these changes were reversed by cholestyramine. In vitro experiments showed that DCA downregulated the differentiation function of ISCs, which was reversed by the AHR agonist 6-formylindolo [3,2-b]carbazole (FICZ). Mechanistically, DCA caused a reduction in indoleamine 2,3-dioxygenase 1 (IDO1) in Paneth cells, resulting in paracrine deficiency of the AHR ligand kynurenine in crypts. We demonstrated for the first time that DCA disrupts intestinal mucosal barrier function by interfering with AHR signalling in ISCs. Supplementation with AHR ligands may be a new therapeutic target for HFD-related impaired intestinal barrier function.

## 1. Introduction

A high-fat diet (HFD) is an important factor in causing metabolic syndrome, type 2 diabetes, non-alcoholic fatty liver disease (NAFLD), and cardiovascular disease [1]. The gut is a complex microenvironment where dietary components, and gut microbes and their metabolites coexist and interact with host cells. The intestine is the organ which is affected the earliest and most directly by high-fat feeding. High fat exposure can lead to the impairment of intestinal barrier function, causing subsequent endotoxaemia and chronic low-grade inflammation [2,3,4], which may be the enterogenic onset of metabolic syndrome. An HFD has been shown to cause elevated levels of bile acids in the gut, particularly secondary bile acids such as deoxycholic acid (DCA) [5,6], which is one of the drivers of impaired intestinal barrier function [7,8,9]. However, the exact mechanism has not been fully elucidated.

The intestinal barrier is maintained by the renewal of intestinal epithelial cells and tight junctions [10,11]. Intestinal stem cells (ISCs) are located in the intestinal crypts and continuously produce a variety of intestinal epithelial cells with different functions, such as secretory cells, including Paneth cells (PCs) and goblet cells (GCs), absorptive enterocytes, and enteroendocrine cells. GCs secrete MUC2 to maintain the integrity of intestinal barrier function [12]. PCs located at the base of the intestinal crypts, as part of the ISC niche, not only secrete a variety of antimicrobial peptides to maintain the balance of the intestinal microenvironment but also provide essential growth factors for ISCs, such as WNT3, EGF, and NOTCH ligands, to maintain their normal function [13]. Thus, the normal proliferation and differentiation functions of ISCs are key factors in maintaining intestinal homeostasis.

Aryl hydrocarbon receptor (AHR) is a ligand-activated transcription factor which is involved in the regulation of physiological activity through environment–cell interactions [14]. The tryptophan component of the diet can be metabolized by indoleamine 2,3-dioxygenase 1 (IDO1) in epithelial cells or immune cells via the kynurenine (KYN) pathway or by intestinal flora to a number of different products, such as indole, indole-3-acetic acid (IAA), or indole-3-aldehyde (IAld) [15,16]. AHR has been confirmed to be expressed in LGR5^+^ ISCs and is involved in regulating the normal proliferation and differentiation of ISCs to secretory cells [17,18,19]. AHR plays important roles in intestinal barrier function and the homeostasis of the intestinal microenvironment. AHR signalling has been reported to maintain intestinal homeostasis by regulating the function of group 3 innate lymphoid cells (ILC3s) and promoting the secretion of IL-22 [20,21,22]. The overexpression of IDO1 in intestinal epithelial cells has been shown to promote the function of the intestinal mucus barrier and ameliorate DSS-induced colitis through activation of AHR signalling [19]. In addition, several studies have linked HFD to decreased AHR signalling, which is an important reason for the impairment of intestinal barrier function as well as the development of metabolic syndrome [23,24].

In this study, we hypothesized that HFD-induced alterations in the bile acid pool, especially an increase in DCA concentrations, interfered with the AHR signalling of ISCs, leading to a reduced differentiation of ISCs to GCs and the subsequent impairment of intestinal barrier function.

## 2. Results

### 2.1. High-Fat Feeding Induces Disturbance of the Bile Acid Pool with an Increased DCA Level

Our previous studies have demonstrated that short-term HFD consumption caused increases in faeces and serum bile acids [25,26,27] and impaired intestinal barrier function [25]; however, the exact composition of faecal bile acids is unknown. Therefore, we used bile acid-targeted metabolomics to examine the faecal bile acid composition in mice fed a low-fat diet (LFD) or an HFD. The results of orthogonal partial least squares discriminant analysis (OPLS-DA) showed that the two groups were well separated (Figure 1A). Compared with the LFD group, faecal total bile acids in the HFD group were significantly increased (Figure 1B), especially the proportion of unconjugated secondary bile acids (Figure 1C). Based on the OPLS-DA model, we found that 10 subtypes of bile acids, including DCA, T-β-MCA, and TCA, had a variable importance of projection (VIP) value > 1, which led to the separation of the two groups (Figure 1D). Heat map and individual bile acid concentrations showed that HFD increased the levels of DCA, TCA, and T-β-MCA compared with the LFD group (Figure 1E,F). In addition, the serum bile acids of mice fed an HFD also increased correspondingly (Figure 1G). These data suggest that two weeks of high-fat feeding rapidly altered the composition of the bile acid pool and, especially, elevated the concentrations of DCA. To investigate the role of cytotoxic DCA in the disruption of intestinal barrier function caused by high-fat feeding, we constructed HFD + cholestyramine (CHO, a bile acid binder) and LFD + DCA groups. The LFD group supplemented with DCA mimicked HFD-induced increases in secondary unconjugated bile acid and DCA levels, whereas concurrent feeding with CHO reversed HFD-induced changes in these bile acid components (Figure 1H,I). In addition, serum total bile acids were significantly increased in the LFD + DCA group, similar to the changes in the HFD group. The administration of CHO reversed HFD-induced bile acid elevation (Figure 1G). Individual bile acid composition ratios in faeces from each mouse are shown in Figure 1J.

### 2.2. High-Fat Feeding-Induced High Levels of DCA Impair the Differentiation of ISCs into GCs

Because a two-week HFD induced a significant increase in cytotoxic DCA concentrations, we next explored the effect of a high level of DCA on intestinal epithelial barrier function. Periodic acid-Schiff (PAS) staining showed that, compared with the LFD group, the number of GCs decreased significantly in both the HFD and LFD + DCA groups, and the administration of CHO prevented the reduction in GCs caused by high-fat feeding (Figure 2A).

Consistent with the histological observations, western blotting and RT-qPCR demonstrated that MUC2 secretion by GCs was significantly reduced in both the HFD and LFD + DCA groups, relative to the LFD group, and cofeeding with CHO reversed the reduction in MUC2 secretion induced by high-fat feeding (Figure 2B,C). These results indicate that high-fat-induced DCA may be a factor in the disruption of intestinal barrier function.

We further investigated whether the reduction in GCs was due to the influence of DCA on the differentiation function of ISCs. Atonal homologue 1 (ATOH1) is a key factor in the differentiation of ISCs to GCs. Consistent with the changes in GCs, we observed that HFD or DCA also significantly reduced the mRNA expression of ATOH1 (Figure 2D). These results suggested that high levels of DCA induced by an HFD led to the impairment of intestinal epithelial barrier function by affecting the differentiation function of ISCs.

### 2.3. High-Fat Feeding-Induced High Levels of DCA Disrupt AHR Signalling in ISCs

Previous studies have demonstrated that proper AHR signalling is a key factor in maintaining the differentiation of ISCs to GCs [17,18]. Therefore, we next explored whether high concentrations of DCA disrupt the function of ISCs by interfering with their AHR signalling mechanism. In morphological studies, we observed that AHR was widely expressed in the gut, which was similar to the results of a study by Park (Figure 3A) [18]. In the crypt regions, AHR was highly expressed between the lysozyme^+^ PCs, in which ISCs were located (Figure 3B). Compared with the results of the control LFD group, there was a significant decrease in the fluorescence intensity of AHR in ileal crypts from mice fed an HFD or a LFD with DCA; however, there was no significant difference in mice fed an HFD with CHO (Figure 3C). To explore the functional changes in AHR, we examined the mRNA and protein expression levels of CYP1A1 downstream of AHR. Similar to the changes in AHR, the CYP1A1 gene and protein expression levels decreased significantly in both the HFD and LFD + DCA groups compared with the LFD group, while there was no significant difference in the HFD + CHO group (Figure 3D,E). These results suggest that a high levels of DCA induced by an HFD may impair the differentiation function of ISCs by interfering with the AHR signalling pathway.

### 2.4. FICZ Restores the Disruption of DCA-Induced AHR Signalling and the Damage to the Differentiation Function of ISCs In Vitro

We further investigated the effect of DCA on the AHR signalling pathway in ISCs in vitro. Ileal tissue and intestinal crypts were cultured in vitro for 24 h with 100 µM DCA or 100 µM DCA with 300 nM 6-formylindolo[3,2-b]carbazole (FICZ), a tryptophan metabolite, acting as a high-affinity ligand for AHR [28]. The mRNA levels of AHR and its downstream CYP1A1 in the isolated intestinal crypt were significantly lower in the DCA-treated group than in the control group, accompanied by reductions in MUC2 and ATOH1 (Figure 4A–D). In addition, DCA administration reduced the expression of MUC2 protein in cultured ileal tissue (Figure 4E).

By administering FICZ to the DCA-treated group, we observed a rebound in CYP1A1 mRNA expression, suggesting that FICZ treatment was able to agitate the AHR signal. FICZ restored the reductions in MUC2 and ATOH1 gene expression mediated by DCA (Figure 4A–D).

To further demonstrate the effects of DCA and AHR on the differentiation function of ISCs, we established intestinal enteroids by isolating intestinal crypts in vitro. After 24 h of in vitro culture, most of the isolated crypts formed enterospheres, and they were then treated with DCA or DCA with FICZ. We observed that, after 72 h of in vitro culture of enteroids, the budding rate of enteroids in the DCA group was significantly lower than that in the control group, and the administration of FICZ partially reversed the decrease in the budding rate that was observed after DCA treatment (Figure 4F). These results suggested that the use of exogenous AHR agonists may be one method to restore the DCA-induced impairment of intestinal barrier function.

### 2.5. An HFD or DCA Diet Induces Changes of the Tryptophan Metabolism in Gut

Tryptophan can be metabolized via the KYN pathway by IDO1 in immune cells or epithelial cells, or be metabolized by intestinal flora via the indole pathway [15,16]. The results of faecal tryptophan-targeted metabolomics showed that, compared with the LFD group, the KYN concentrations decreased significantly in both the HFD and LFD + DCA groups (Figure 5A). Although the measured faecal tryptophan concentrations in the LFD + DCA group were higher than those in the HFD group, the IDO1 activity (KYN/tryptophan) in both groups was significantly lower than in the LFD group (Figure 5B,C), which is consistent with the observed changes in AHR signalling. In addition, we observed no significant changes in the main tryptophan metabolites of bacteria IAA, indole, or IAld in both the HFD and LFD + DCA groups compared with the LFD group (Figure 5D–F).

### 2.6. DCA Downregulates IDO1 in PCs in Intestinal Crypts

A previous study illustrated the presence of IDO1^+^ PCs in intestinal crypts [29]. PCs are located in the intestinal crypts and adjacent to ISCs, and play a protective and supportive role for ISCs [13]. To clarify the mechanism of HFD/DCA with regards to the AHR signalling pathway of ISCs, we further explored the expression of IDO1 in crypt cells. The immunological results showed that IDO1 was expressed in the crypt regions with low expression of AHR and positive lysozyme staining, which corresponds to the position of PCs (Figure 6A,B). IDO1^+^ PCs (white dashed line) and IDO1^−^ PCs (white line) were presented in crypts (Figure 6B). Compared with the LFD group, the IDO1 fluorescence intensity in the crypt regions decreased significantly in both the HFD and LFD + DCA groups (Figure 6C). The IDO1 protein and gene expression results confirmed the decreases in the HFD and LFD + DCA groups (Figure 6D,E).

### 2.7. DCA Downregulates IDO1 In Vitro

In addition, we also found that IDO1^+^ PCs (white arrow) and IDO1^−^ PCs (white triangles) existed in enteroids (Figure 7A). Enteroids also showed less IDO1 expression in response to DCA stimulation (Figure 7B). In vitro, isolated crypt cultures also showed that direct stimulation by DCA resulted in a downregulation of IDO1 mRNA levels in crypts (Figure 7C). These results indicate that high levels of DCA induced by an HFD may cause damage to AHR signalling in ISCs by downregulating IDO1 expression in crypt regions.

## 3. Discussion

In this study, we observed that a two-week HFD resulted in a significant increase in cytotoxic DCA. Elevated DCA rapidly affected tryptophan metabolism in the crypt regions by downregulating IDO1 and further altered the differentiation function of ISCs through AHR signalling, leading to the reduction of GCs and the weakening of mucus barrier function.

Chronic HFD consumption can lead to disturbances in the bile acid pool, which may further contribute to the development of metabolic syndrome [30] and colorectal cancer [31]. Through a bile acid-targeted metabonomics analysis, we found that even two weeks of HFD caused significant changes in the bile acid pool. The HFD induced significant increases in faecal concentrations of unconjugated secondary bile acids, especially cytotoxic DCA. Previous studies have shown that high-fat feeding doubles bile acid levels and that a similar level of bile acid can be achieved by adding 0.2% DCA to the diet [9,27,32]. In our study, 0.2% DCA supplementation significantly increased faecal DCA concentrations in mice fed a LFD, while concurrent feeding with CHO reversed the increase in DCA induced by an HFD. Therefore, CHO can be used as an appropriate intervention factor for DCA.

HFD-induced impairment of intestinal barrier function may be associated with elevated bile acid levels [33]. Bile acid can signal through its receptors, farnesoid X receptor (FXR) or Takeda G-protein-coupled receptor 5 (TGR5), to maintain the homeostasis of the intestinal epithelium [10,34]. However, excessive bile acids, especially cytotoxic DCA, have been proven to destroy intestinal barrier function [7,8,9] and promote colorectal cancer progression by inducing oxidative stress, inhibiting FXR signalling, or activating Wnt and EGFR signalling [29,35,36,37,38]. Several studies have demonstrated that DCA induces the downregulation of intestinal tight junction proteins, such as ZO-1 and occludin, and reduces the number of GCs [7,8,9,26,27]. In addition, our team has confirmed that the DCA-induced dysfunction of PCs leads to gut dysbiosis [25]. In the present study, we observed that an HFD or DCA diet induced decreases in the number of GCs and the secretion of MUC2, probably due to insufficient differentiation of ISCs to GCs. In contrast, the administration of CHO along with an HFD to mice resulted in the normalization of GC and MUC2 secretion, similar to mice receiving a LFD.

The environmental sensor AHR is widely expressed in the gut. Proper AHR signalling plays a critical role in maintaining the integrity of intestinal barrier function [39,40]. AHR mediates IL-22 production in ILC3s, which maintains intestinal homeostasis and resists colonization by the pathogenic microorganism *Candida albicans* [20,21,22]. In addition, the regulation of ISCs by the ligand-dependent activation of AHR has been demonstrated in recent years. Studies have demonstrated that AHR signalling maintains normal ISC proliferation by promoting Znrf3 and Rnf43 expression to suppress aberrant WNT-β-catenin signalling and tumour progression [17,18]. The regulation of AHR signalling in ISCs includes the promotion of GC differentiation and the maintenance of intestinal barrier function. The loss of AHR signalling can lead to attenuation of MUC2 expression [17,41]. In the present study, we found that AHR signalling in the crypt regions was attenuated in HFD- or DCA-fed mice and could be reversed by the concurrent administration of CHO. Supplementation with exogenous AHR ligands, such as dietary indole-3-carbinol (I3C) or the high-affinity AHR ligand IAld producer *Lactobacillus royale*, has been shown to partially antagonize the intestinal barrier function impairment associated with an HFD [23,42]. In addition, FICZ, a potent agonist of AHR, has also shown potential therapeutic effects on the high-fat or DSS-induced disruption of intestinal barrier function in in vivo and in vitro experiments [23,27]. Our experiments also confirmed that the in vitro administration of FICZ partially reversed the detrimental effect of DCA on the differentiation function of ISCs and restored MUC2 expression in the intestinal mucosa. In addition, FICZ partially reversed the DCA-induced reduction in the budding of enteroids. This evidence suggests that DCA can affect intestinal homeostasis via the AHR pathway, further suggesting that exogenous supplementation with CHO or AHR agonists may be used as an alternative treatment for HFD-induced impairment of intestinal barrier function.

Approximately 95% of dietary tryptophan is metabolized in the gut by IDO1 in epithelial and immune cells via the KYN pathway, and some is metabolized by intestinal flora to indoles and its derivatives or by the serotonin pathway in enterochromaffin cells [15,16,43]. Patients with a chronic HFD or metabolic syndrome tend to have lower faecal AHR activity than healthy individuals, which correlates strongly with an unfavourable metabolic phenotype [23,24]. These studies have attributed this phenomenon to the excessive plundering of tryptophan by inflammation-induced high IDO1 activity, which reduces flora-derived highly active AHR ligands such as IAA and indole and disrupts intestinal barrier function by reducing downstream IL-22 signalling [23,24]. However, our study showed that, unlike its long-term effects, a two-week high-fat feeding or DCA feeding decreased faecal KYN concentrations and IDO1 activity, without altering the bacterial metabolites IAA, indole, or IAld. Therefore, we considered that a short-term HFD may mainly affect AHR signalling via the KYN pathway of tryptophan metabolism.

Morphological studies revealed IDO1 expression in the intestine [44], and Sandra and colleagues observed the presence of IDO1^+^ PCs in the intestinal crypts, especially in the distal small intestine [29], which is consistent with our observations. IDO1^+^ PCs may lead to the excessive consumption of tryptophan and greater KYN production in colorectal cancer crypts, thereby inhibiting the cell cycle of CD8^+^ T cells and the expansion of immunosuppressive Tregs [29]. We hypothesized that local tryptophan metabolism in the terminal ileum crypts may be mainly regulated by IDO1 expressed in PCs, then regulates AHR activity in ISCs through KYN paracrine signalling. The present study showed that high-fat feeding or DCA feeding decreased KYN production by downregulating IDO1 expression in crypts and IDO1 activity in faeces, which was consistent with the changes observed in AHR signalling. Organoid studies further confirmed that the IDO1^+^ cell number in the enterosphere was downregulated by DCA treatment. These results suggest that IDO1-KYN may be the main upstream of AHR signalling in ISCs.

Several studies have shown that the IDO1-KYN-AHR axis plays a role in the induction of an immunosuppressive microenvironment in tumours [45,46]; however, its role in promoting intestinal homeostasis and regulating inflammatory responses should not be ignored [19,47]. Enteroids from IDO1 transgenic mice showed a higher secretory lineage gene, such as ATOH1, Gfi1, or MUC2, while enteroids from IDO1 knockout mice showed a lower level than wide-type mice, suggesting that IDO1 maintained the function of intestinal secretory cell differentiation. Mechanically, inflammation-induced IDO1 expression can regulate the paracrine effect of KYN and, together with the nonenzymatic agonist effect of AHR, it participates in regulating the differentiation of secretory cells and improves the intestinal mucus barrier [19]. KYN is a moderate-affinity ligand of AHR and is sufficient to activate the transcriptional activity of genes downstream of AHR [48,49]. Park and colleagues observed that tryptophan and KYN promoted the differentiation of GCs in HT-29 cells, and IDO1 inhibitor 1-Methlytryptophan (1-MT) and AHR inhibitor α-naphthoflavone inhibited the gene expression of MUC2 [50]. Therefore, an HFD-induced elevation of DCA can lead to disorder of the IDO1-KYN-AHR axis in the crypt regions, which can affect mucosal barrier function and the differentiation of ISCs into GCs in a short period of time.

This study still has some limitations. Mice have a different bile acid metabolic profile than humans. For example, there is a greater variety of primary bile acid in mice (CA, CDCA, UDCA, α-MCA, and β-MCA) than in humans (CA and CDCA), resulting in consequent differences in intestinal secondary bile acids [51,52]. In addition, in mice, most bile acids are conjugated to taurine, while in humans, bile acids are conjugated to either glycine or taurine [53]. Therefore, in further experiments, we will work on analysing whether a short-term HFD brings about similar changes in humans. In addition, to better explore the mechanisms of IDO1 and AHR in a short-term HFD-induced disruption of intestinal barrier function, the use of transgenic mice targeting intestinal mucosal IDO1 or AHR would be a better option in the future.

In conclusion, our study demonstrates that a high level of cytotoxic DCA induced by an HFD reduced the levels of tryptophan metabolism to KYN by inhibiting the activity of IDO1 in PCs. The reduction in KYN production resulted in insufficient activation of the AHR signalling, disrupting the differentiation of ISCs to GCs. Therefore, targeting elevated DCA levels or exogenous supplementation with AHR ligands may be potential therapeutic options for ameliorating the HFD-induced disruption of intestinal barrier function.

## 4. Materials and Methods

### 4.1. Mice and Diets

Six-week-old male C57BL/6J mice (18–22 g) were purchased from the Animal Center of Shanghai General Hospital. After one week of adaptive feeding on normal chow, mice were randomly divided into four groups and fed for 2 consecutive weeks with (i) a control LFD (10 kcal% fat, Trophic Animal Feed High-Tech Co., Ltd., Nantong, China), (ii) an HFD (60 kcal% fat, Trophic Animal Feed High-Tech Co., Ltd., Nantong, China), (iii) an HFD + CHO diet (HFD mixed with 6% (*w*/*w*) CHO, C4650, Sigma-Aldrich, Saint Louis, MO, USA), or (iv) a LFD + DCA diet (LFD mixed with 0.2% DCA, D2510, Sigma-Aldrich).

In a separate group, mice were randomly divided into three groups and fed the following diets for 2 consecutive weeks: (i) a LFD (10 kcal% fat), (ii) an HFD (60 kcal% fat), or (iii) a LFD + DCA diet (HFD mixed with 0.2% DCA).

After a 2-week period, these mice were food-deprived for 12 h and then euthanized. The ileal tissues, fresh faeces, and serum were harvested and stored at −80 °C for further studies. All of these studies were approved by the Institutional Animal Care and Use Committee of Shanghai General Hospital.

### 4.2. Measurement of Faecal Bile Acid, KYN and Tryptophan Concentrations

Faeces were collected from mice for the quantitative determination of bile acids, KYN, and tryptophan. The bile acid and tryptophan concentrations in faeces were measured by liquid chromatography-tandem mass spectrometry (LC-MS/MS) [54,55,56]. Faeces were mixed with 400 µL of methanol and vortexed for 60 s, followed by grinding with glass beads in a high-throughput tissue grinder at 55 Hz for 60 s, which was repeated three times. After sonication for 30 min at room temperature, a series of centrifugations, and the addition of methanol and vortexing, were carried out, followed by the supernatant being taken for testing. The mass spectrometer was operated in negative mode using electrospray ionization. A standard curve was established from the concentrations and peak areas of the standard solutions, and the concentrations of KYN, tryptophan, and different components of bile acids were calculated from the regression equations.

### 4.3. Measurement of Serum Bile Acid

Mice were anaesthetized by chloral hydrate, and fresh mouse serum was obtained by blood sampling from the eyes. Then, the mice were killed by exsanguination. The total bile acid concentrations in the serum was measured using an enzymatic assay kit (E003-2-1, Nanjing Jiancheng Biological Engineering Institute, Nanjing, China). The enzymatic cycling method was used to analyse the concentrations of serum total bile acid [57]. An enzymatic cycling reaction system was constructed for both the standard and tested samples. After incubation at 37 °C for 1 min, the absorbance at 405 nm (A0) was read, and then the absorbance at 405 nm (A1) was read after incubation at 37 °C for 3 min. The absorbance changes (A1 − A0) in the standard and tested samples were calculated. The total bile acid concentrations of the tested sample was obtained by the following equation: [(A1tested sample − A0tested sample)/(A1standard sample − A0standard sample)] × standard sample concentrations.

### 4.4. Periodic Acid-Schiff Staining

The terminal ileal tissue was fixed with 4% paraformaldehyde for 24 h, subsequently embedded in paraffin, and cut into 4 µm slices. PAS staining was performed for the quantitative analysis of GCs. Five fields were counted for each mouse, and the average number of GCs of each villi-crypt unit was recorded.

### 4.5. Immunoblot Analysis

Protein in ileal tissues was extracted by RIPA buffer (EpiZyme, Shanghai, China) with a protease and phosphatase inhibitor (Beyotime, Nanjing, China). The tissues were homogenized in a high-throughput tissue grinder at 70 Hz for 60 s, which was repeated twice. The mixed solution was placed on ice for 1 h and then centrifuged at 12,000 rpm for 10 min. SDS buffer was added to the supernatant at a 4:1 ratio and then heated at 100 °C for 10 min. Equal amounts of protein in 7.5% or 10% SDS-PAGE gels were used for electrophoresis and then transferred to polyvinylidene difluoride membranes (Millipore, Tullagreen, Ireland). Five percent nonfat milk was used to block nonspecific binding of membranes, which were then probed with primary antibodies against MUC2 (1:1000, A14659, Abclonal, Wuhan, China), CYP1A1 (1:1000, 13241-1-AP, Proteintech, Wuhan, China), IDO1 (1:1000, D8W5E, Cell Signaling Technology, Danvers, MA, USA), β-actin (1:1000, AF7018, Affinity Biosciences, Shanghai, China), and GAPDH (1:5000, 10494-1-AP, Proteintech) at 4 °C overnight. On the second day, the membranes were incubated with HRP-conjugated goat anti-rabbit antibody (1:100,000, Jackson ImmunoResearch, Baltimore, MD, USA) for 1 h. The ECL chemiluminescence method was used to visualize the target bands, and ImageJ software (version 1.53e) was used to analyse the target bands.

### 4.6. RNA Extraction and RT-qPCR

Total RNA from mouse ileal tissues and isolated intestinal crypts was extracted by TRIzol (Takara, Shiga, Japan). RNA (1000 ng) was reverse-transcribed to cDNA using a HyperScript III RT SuperMix kit (EnzyArtisan, Shanghai, China). The target primer was then quantified by real-time qPCR analysis, which was performed in a 10 µL system with a Universal SYBR qPCR Mix kit (EnzyArtisan, Shanghai, China). The primer sequences are shown in Table 1. The mRNA expression levels of the target genes were normalized to those of GAPDH and quantified using the 2-ΔΔct method.

### 4.7. Isolation of Crypts and Culture of Enteroids and Crypts In Vitro

After sacrifice of the mice, the terminal ileal tissue (approximately 4–5 cm) was removed and immediately rinsed in ice-cold PBS and longitudinally opened. Then, the tissue was cut into 0.5 cm sections and thoroughly flushed with ice-cold PBS. The tissue fragments were then incubated in 3 mM ethylenediaminetetraacetic acid (EDTA) solution at 4 °C for 30 min. After removing the EDTA solution, the tissue fragments were shaken vigorously approximately 10 times in a 15 mL tube using ice-cold PBS and filtered through a 70-µm filter to obtain small intestinal crypts. The samples were centrifuged at 250× *g* for 10 min at 4 ℃.

The sediment at the bottom of the tube was the Isolated crypts. After counting the crypts under the microscope, they were diluted to 100/10 µL with PBS buffer, and then 20 µL of the mixture was mixed with 20 µL of Matrigel (356231, Corning, NY, USA) and inoculated into 48-well plates. The samples were cultured at 37 °C for 15 min, followed by the addition of medium (IntestiCult™ Organoid Growth Medium, STEMCELL Technologies, Vancouver, BC, Canada).

After 24 h, enteroids were exposed to DMSO, 100 μM DCA (D2510, Sigma-Aldrich), and DCA + FICZ (BML-GR206-0100, Enzo Life Sciences, Farmingdale, NY, USA) for 48 h. The medium was replaced 48 h after plating. After culture, the budding rate of 50 enteroids per well was counted at 72 h by light microscopy.

The remaining crypts were cultured for RT-qPCR. Crypts were placed in a six-well plate containing 5 mL of DMEM (containing 10% foetal bovine serum and 100 U/mL penicillin-streptomycin double antibody solution). The experiment was divided into three groups: (i) a control (NC) group, treated with the same volume of DMSO as groups (ii) and (iii); (ii) a DCA group, treated with DCA at a concentration of 100 µM; (iii) a DCA + FICZ group, treated with DCA at a concentration of 100 µM and FICZ at a concentration of 300 nM. The crypts were incubated at 37 °C in an incubator with 5% CO2. After 24 h, the intestinal crypts were collected for subsequent RT-qPCR.

### 4.8. Ileal Tissue Culture In Vitro

The terminal ileal tissue was removed after dissecting the mice, and then the tissue was cut into 0.5 cm sections and placed in six-well plates containing 10 mL of DMEM (containing 10% foetal bovine serum and 100 U/mL penicillin-streptomycin double antibody solution) for 24 h. The experiment was divided into three groups: (i) a control (NC) group, treated with the same volume of DMSO as groups (ii) and (iii); (ii) a DCA group, treated with DCA at a concentration of 100 µM; (iii) a DCA + FICZ group, treated with DCA at a concentration of 100 µM and FICZ at a concentration of 300 nM. The tissues and crypts were incubated at 37 °C in an incubator with 5% CO_2_ and changed once every 12 h. After 24 h, ileal tissues were collected for subsequent Western blotting.

### 4.9. Immunofluorescence Staining

The ileal tissue was fixed for 24 h using 4% paraformaldehyde, embedded in paraffin, and subsequently sectioned. After a series of dewaxing, hydration, and antigen repair processes, tissue sections were subjected to blocking buffer (Beyotime, Nanjing, China) for 1 h and subsequently stained. The enteroids were cultured in the cell climbing slice. They were fixed for 1 h using 4% paraformaldehyde and then subjected to blocking buffer for 1 h at room temperature. Primary antibodies against AHR (1:100, MA1-513, Invitrogen, Waltham, MA, USA), IDO1 (1:1600, D8W5E, Cell Signaling Technology), and lysozyme (1:200, A0099, Dako or 1:100, ab36362, Abcam, Cambridge, UK) were used for staining for 16 h at 4 °C. After rewarming for 1 h at room temperature, the tissue sections were incubated with secondary antibodies (1:200, Yeasen, Shanghai, China) for 1 h at room temperature. Then, the sections were incubated with DAPI (Yeasen, Shanghai, China) and imaged under a fluorescence microscope. The images were captured by a Leica DMI8 and a Leica STELLARIS 8 DIVE. Images were analysed by ImageJ software (version 1.53e).

### 4.10. Statistical Analysis

Data are shown as the mean ± SEM, with the number of mice indicated. Statistical significance was determined by GraphPad Prism (version 9.0.2). Metabolomics data was analyzed by R software (version 3.5) and SIMCA-P 14.1 (Umetrics, Umeå, Sweden). The parameter test was conducted by unpaired *t* test or one-way ANOVA. A *p* value less than 0.05 was considered statistically significant.

## Figures and Tables

**Figure 1 ijms-23-15578-f001:**
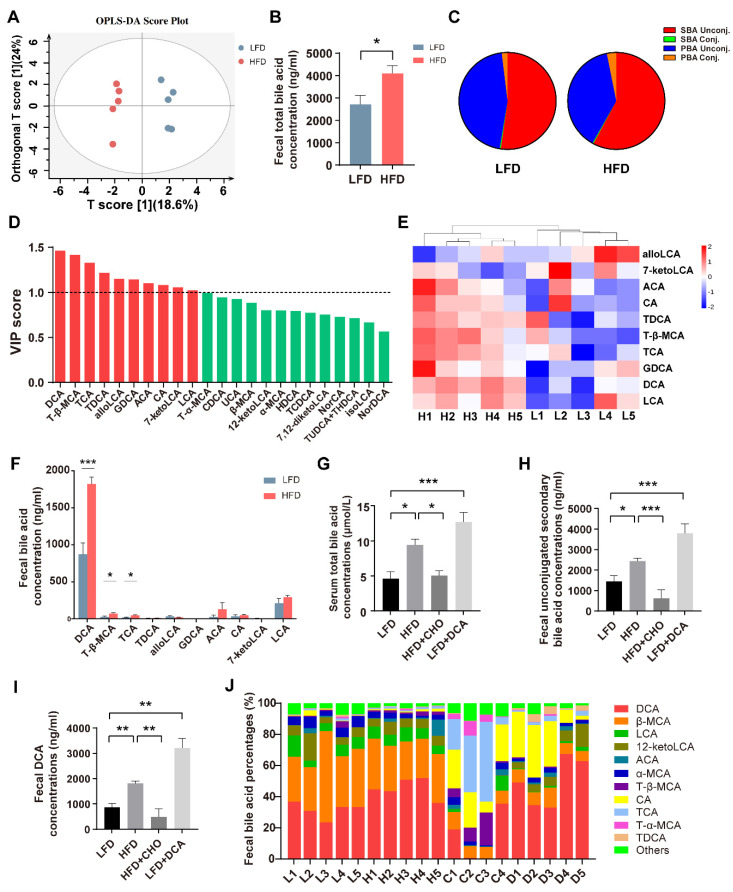
High-fat feeding induces disorder of the bile acid pool and increases DCA levels. (**A**) Orthogonal partial least squares discriminant analysis (OPLS-DA) of faeces in the LFD and HFD groups. R2Y = 0.983, Q2 = 0.647. (**B**–**E**) Comparison of faecal bile acids in total bile acid concentrations (**B**), the bile acid composition ratio (**C**), variable importance of projection (VIP) value (red bars: VIP > 1) (**D**), heat map of bile acids with VIP > 1 (**E**), and individual bile acid concentrations (**F**) in the LFD and HFD groups (n = 5). (**G**) Serum total bile acid concentrations in the LFD, HFD, HFD + CHO, and LFD + DCA groups (n = 5). (**H**–**J**) Faecal bile acid in secondary unconjugated bile acid concentrations (**H**), DCA concentrations (**I**), and the bile acid composition ratio (**J**) in the LFD, HFD, HFD + CHO, and LFD + DCA groups (n = 4–5). Data are shown as the mean ± SEM. Unconj., unconjugated; Conj., conjugated; LFD, low-fat diet; HFD, high-fat diet; DCA, deoxycholic acid; CHO, cholestyramine; L, LFD; H, HFD; C, HFD + cholestyramine; D, LFD + deoxycholic acid. * *p* < 0.05; ** *p* < 0.01; *** *p* < 0.001.

**Figure 2 ijms-23-15578-f002:**
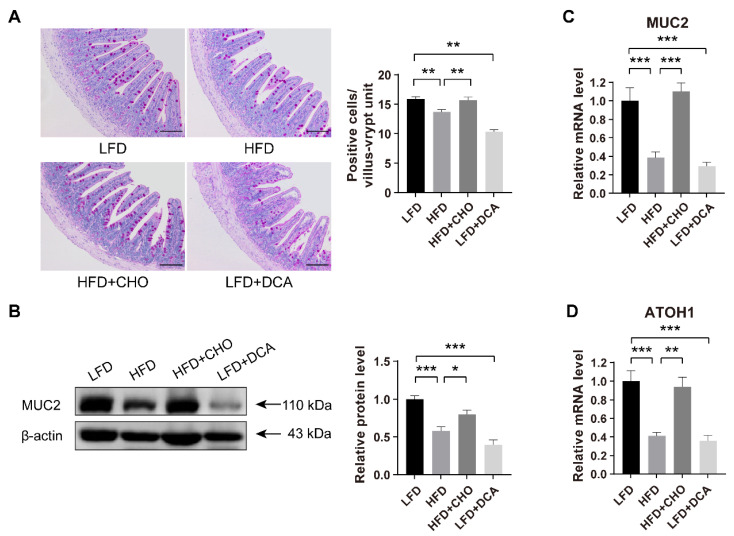
Elevated DCA induced by an HFD impairs the differentiation of ISCs to GCs. (**A**) PAS staining and analysis of positive cells per villus–crypt unit (n = 5); scale bars, 100 μm. (**B**) Relative protein levels of MUC2 secreted by GCs in ileal tissue and analysis in different groups (n = 6). (**C**) Relative mRNA levels of MUC2 in ileal tissue and analysis in different groups (n = 6). (**D**) The mRNA expression levels of ATOH1, the key gene of ISC differentiation into GCs in the ileum, in different groups (n = 6). Data are shown as the mean ± SEM. PAS, periodic acid-schiff; ISCs, intestinal stem cells; GCs, goblet cells; ATOH1, atonal homologue 1; LFD, low-fat diet; HFD, high-fat diet; CHO, cholestyramine; DCA, deoxycholic acid. * *p* < 0.05; ** *p* < 0.01; *** *p* < 0.001.

**Figure 3 ijms-23-15578-f003:**
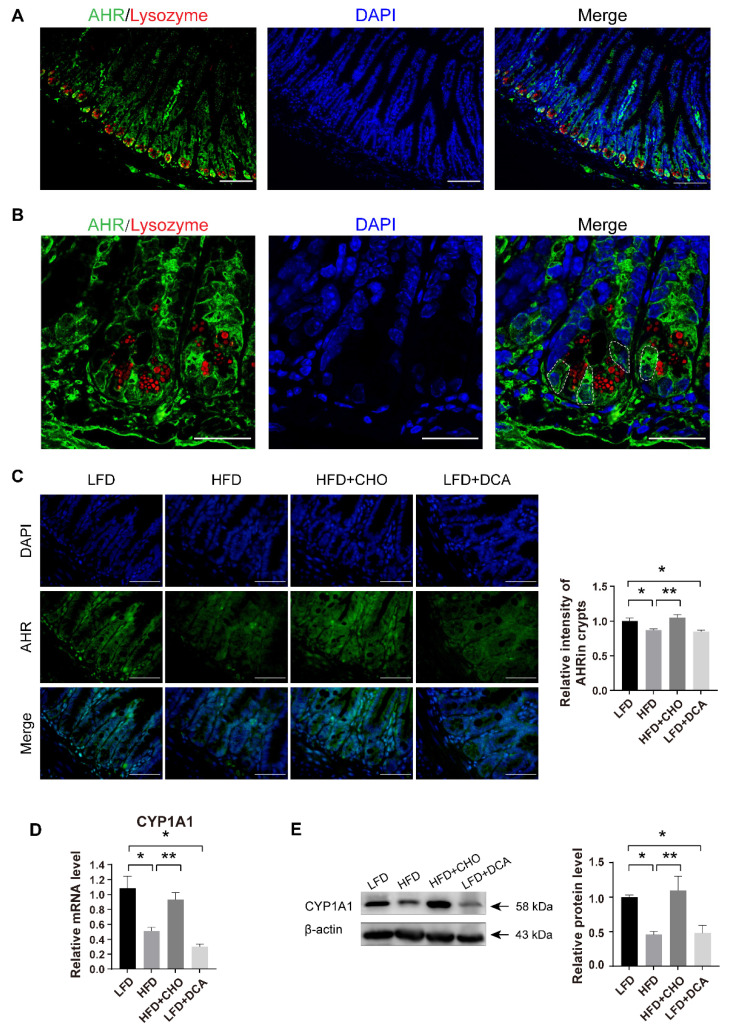
Elevated DCA induced by an HFD disrupts AHR signalling in ISCs. (**A**,**B**) Immunofluorescence staining images of DAPI (blue), AHR (green) and the Paneth cell maker lysozyme (red) in ileal tissue (**A**) and ileal crypts (**B**). The white dashed line indicates the location of the ISCs. Scale bars, 100 μm (**A**) and 25 μm (**B**). (**C**) Representative immunofluorescence images of AHR and analysis of fluorescence intensity in different groups (n = 5); scale bars, 100 μm. (**D**) The mRNA expression levels of the AHR downstream gene CYP1A1 in the ileum in different groups (n = 4). (**E**) Relative protein levels of CYP1A1 in ileal tissue and analysis in different groups (n = 5). Data are shown as the mean ± SEM. n = 5. AHR, aryl hydrocarbon receptor; ISCs, intestinal stem cells; LFD, low-fat diet; HFD, high-fat diet; CHO, cholestyramine; DCA, deoxycholic acid. * *p* < 0.05; ** *p* < 0.01.

**Figure 4 ijms-23-15578-f004:**
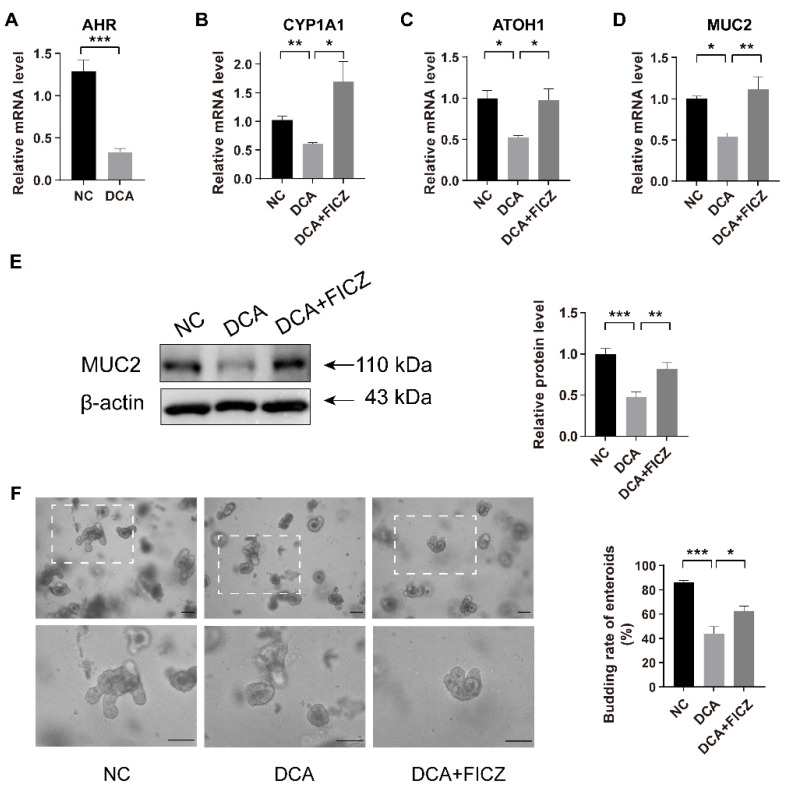
Supplementation with AHR agonists rescues DCA-induced damage to ISCs differentiation. (**A**–**D**) Relative mRNA expression levels of AHR (**A**), CYP1A1 (**B**), ATOH1 (**C**), and MUC2 (**D**) in isolated crypts cultured with different administrations for 24 h (n = 4). (**E**) Relative protein levels and analysis of MUC2 in isolated ileal tissue cultured with different treatments for 24 h (n = 6). (**F**) Representative images of intestinal enteroids and analysis of enteroid budding rates after culturing for 72 h in different groups (n = 3); scale bars, 100 μm. Data are shown as the mean ± SEM. AHR, aryl hydrocarbon receptor; ISCs, intestinal stem cells; ATOH1, atonal homologue 1; NC, negative control; DCA, deoxycholic acid; FICZ, 6-formylindolo[3,2-b]carbazole. * *p* < 0.05; ** *p* < 0.01; *** *p* < 0.001.

**Figure 5 ijms-23-15578-f005:**
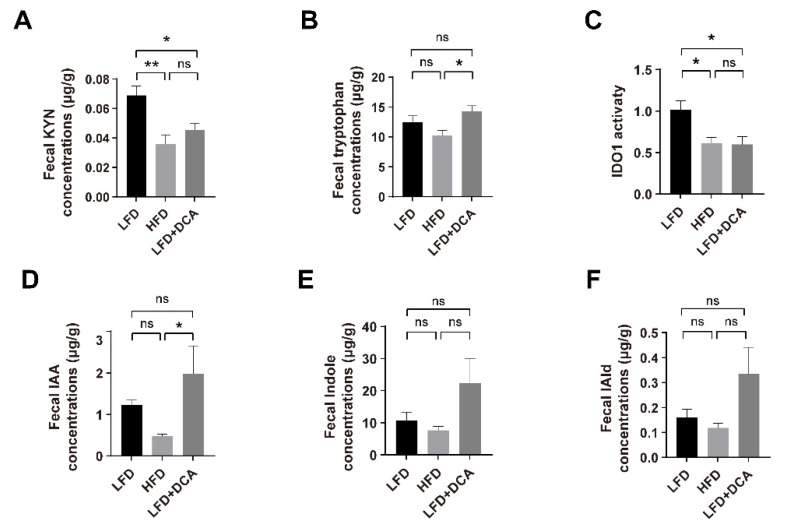
An HFD or DCA diet induces changes in the tryptophan metabolism. (**A**–**F**) Faecal KYN concentrations (**A**), tryptophan concentrations (**B**), IDO1 activity (KYN/tryptophan) (**C**), IAA concentrations (**D**), Indole concentrations (**E**), and IAld concentrations (**F**) in different groups (n = 5). Data are shown as the mean ± SEM. KYN, kynurenine; IDO1, indoleamine 2,3-dioxygenase 1; IAA, indole-3-acetic acid; IAld, indole-3-aldehyde. * *p* < 0.05; ** *p* < 0.01; ns, no significant difference.

**Figure 6 ijms-23-15578-f006:**
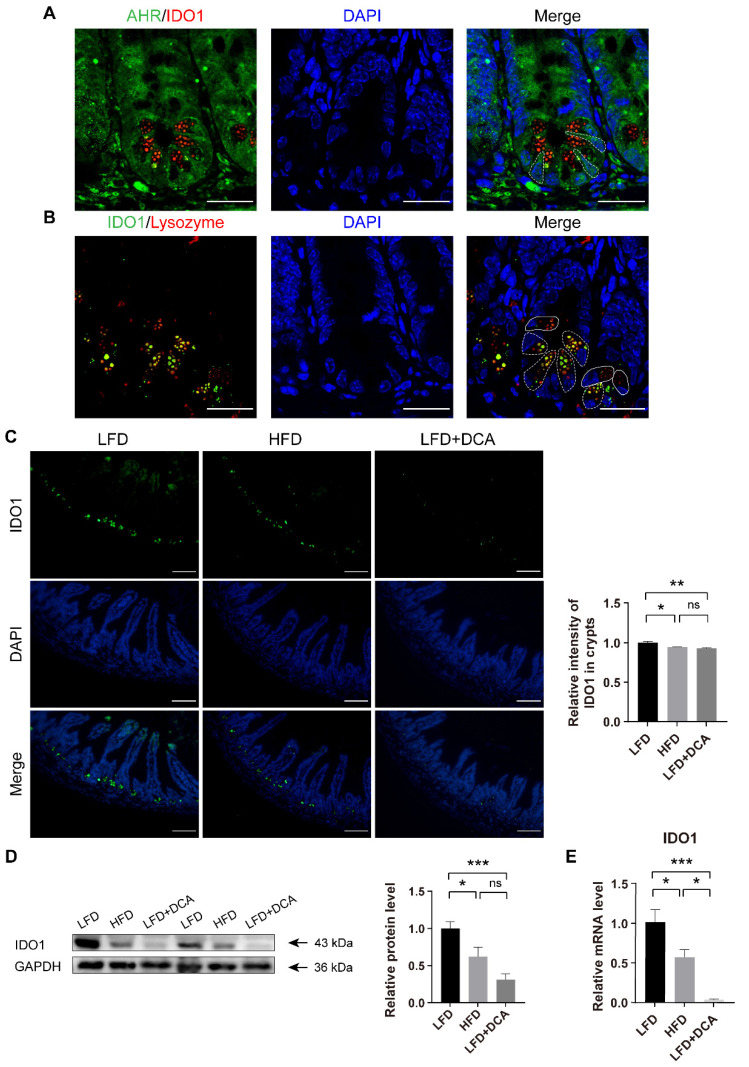
DCA downregulates IDO1 in PCs in intestinal crypts. (**A**) Immunofluorescence staining of DAPI (blue), lysozyme (red) and IDO1 (green) showed that IDO1 was expressed in low-AHR fluorescence intensity areas in ileal crypts. The white dashed line indicates the location of the ISCs; scale bars, 25 μm. (**B**) Immunofluorescence staining of DAPI (blue), lysozyme (red) and IDO1 (green) showed that IDO1 was expressed in PCs. The white dashed line indicates IDO1^+^ PCs, and the white line indicates IDO1^−^ PCs; scale bars, 25 μm. (**C**) Representative immunofluorescence images of IDO1 and analysis of fluorescence intensity in different groups (n = 5); scale bars, 100 μm. (**D**) Relative protein levels of IDO1 in ileal tissue and analysis in different groups (n = 7). (**E**) Relative mRNA levels of IDO1 in ileal tissue (n = 5) in different groups. Data are shown as the mean ± SEM. PCs, Panteh cells; AHR, aryl hydrocarbon receptor; IDO1, indoleamine 2,3-dioxygenase 1; LFD, low-fat diet; HFD, high-fat diet; DCA, deoxycholic acid. * *p* < 0.05; ** *p* < 0.01; *** *p* < 0.001; ns, no significant difference.

**Figure 7 ijms-23-15578-f007:**
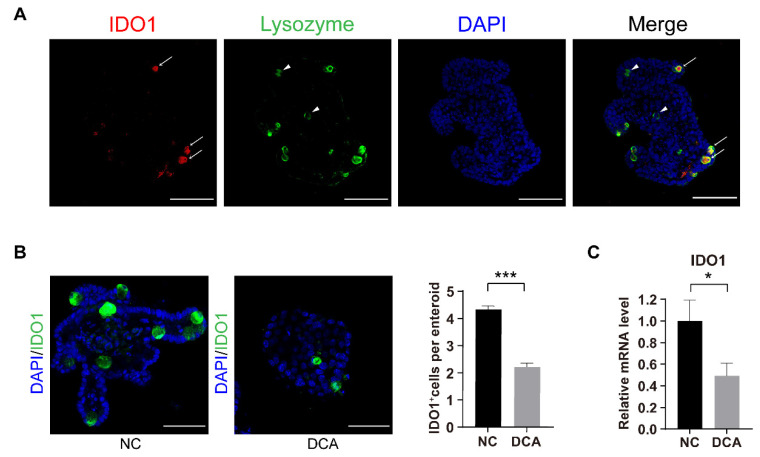
DCA downregulates IDO1 in vitro. (**A**) Immunofluorescence staining of DAPI (blue), IDO1 (red) and lysozyme (green) in enteroids. White arrow, IDO1^+^ PCs; white triangles, IDO1^−^ PCs; scale bars, 100 μm. (**B**) Representative immunofluorescence images of IDO1 in enteroids cultured with DMSO or 100 μM DCA for 48 h and relative analysis of IDO1^+^ cells per enteroid (n = 3); scale bars, 25 μm. (**C**) Relative mRNA levels of IDO1 in isolated crypts cultured in vitro (n = 7) in different groups. Data are shown as the mean ± SEM. IDO1, indoleamine 2,3-dioxygenase 1; PCs, Paneth cells; NC, negative control; DCA, deoxycholic acid. * *p* < 0.05; *** *p* < 0.001.

**Table 1 ijms-23-15578-t001:** The primer sequence used for the evaluation of mRNA sequence of this article.

Gene (Mouse)	Primer Sequence (5′ to 3′)
*GAPDH*-F	CATCACTGCCACCCAGAAGACTG
*GAPDH*-R	ATGCCAGTGAGCTTCCCGTTCAG
*AHR*-F	CTGGTTGTCACAGCAGATGCCT
*AHR*-R	CGGTCTTCTGTATGGATGAGCTC
*ATOH1*-F	CCTTCAACAACGACAAGAAGCTG
*ATOH1*-R	GCAACTCCGACAGAGCGTTG
*CYP1A1*-F	ACCCTTACAAGTATTTGGTCGT
*CYP1A1*-R	GTCATCATGGTCATAACGTTGG
*MUC2*-F	AAACCTCCAACTGAATCCTCG
*MUC2*-R	GAAGTGACGAATGGTGATGTTG
*IDO1*-F	GGTCTCTGTGAGAAAGTTCCACCTC
*IDO1*-R	AGTCCCTCTGCTTTCCACATTTG

F-forward sequence; R-reverse sequence.

## Data Availability

The data presented in the study are included in the article. Further inquiries can be directed to the corresponding authors.

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
