# Peer review of "Intestinal Stem Cells Damaged by Deoxycholic Acid via AHR Pathway Contributes to Mucosal Barrier Dysfunction in High-Fat Feeding Mice"

_ijms, 2022, doi:10.3390/ijms232415578_

Round 1
Reviewer 1 Report
Thank you for author's great work and submit your precious science results in IJMS.
Authors demonstrates that 1) HFD-induced bile acid pool issue with DCA increase, HFD-induced DCA impairs 2) intestinal cell differentiation through 3) AHR signaling in ISCs. 4) HFD or DCA changes tryptophan metabolism in the intestine via IDOI in Paneth cells in vivo and in vitro.
Overall, author's manuscript is good for the publication and it is beneficial to share and discuss in IJMS for the contribution sciences.
I just want authors manifest some points for better paper;
1. Please present the information of mouse sex or justification for a specific sex.
2. In Figure 1, please present better image (some panel has not clear x or y axis).
3. In result 2.4, I think it is better to define or explain FICZ, agonist of AHR.
4. Could you refer 1 or 2 references for different bile acid metabolic profile between human and mice? (Discussion).
Thank you!
Author Response
Response to Reviewer 1 Comments
Point 1: Please present the information of mouse sex or justification for a specific sex.
Response 1: Thanks for the reviewer’s suggestions. The sex information of mouse has been added on page 15, line 396.
Point 2: In Figure 1, please present better image (some panel has not clear x or y axis).
Response 2: According to the reviewer’s suggestions, we have revised some panels and improved the images in Figure 1.
Point 3: In result 2.4, I think it is better to define or explain FICZ, agonist of AHR.
Response 3: We accept the proposal. The defination of FICZ has been added in section 2.4.
Point 4: In Figure 1, please present better image (some panel has not clear x or y axis).
Response 4: Thanks for the reviewer’s suggestions. In the revised manuscript we included the differences for the bile acid metabolic profile between human and mouse and related references in the Discussion (page 14, line 379-383).

Reviewer 2 Report
The authors presented one very interesting study entitled “Intestinal stem cells damaged by deoxycholic acid via AHR 2 pathway contributes to mucosal barrier dysfunction in high-fat 3 feeding mice”.
The detailed study analysed the influence of high fat diet in the intestinal barrier function.
All the experiments are clearly described and the hypothesis are supported by the results, with mRNA transcription levels, protein levels, histological nalysyis, and immunofluorescence thecnics.
They proved the hypothesis that HFD-induced alterations in the bile acid pool, especially the increase in DCA concentration, interfered with the AHR signalling of ISCs, and proposed exogenous supplementation with AHR ligands as one promissing therapeutic procedure to combat the intestinal cells damage.
Author Response
Response to Reviewer 2 Comments
Point: The authors presented one very interesting study entitled “Intestinal stem cells damaged by deoxycholic acid via AHR 2 pathway contributes to mucosal barrier dysfunction in high-fat 3 feeding mice”.
The detailed study analysed the influence of high fat diet in the intestinal barrier function.
All the experiments are clearly described and the hypothesis are supported by the results, with mRNA transcription levels, protein levels, histological nalysyis, and immunofluorescence thecnics.
They proved the hypothesis that HFD-induced alterations in the bile acid pool, especially the increase in DCA concentration, interfered with the AHR signalling of ISCs, and proposed exogenous supplementation with AHR ligands as one promissing therapeutic procedure to combat the intestinal cells damage.
Response : We so appreciate reviewer’s comments.
